# LAPIS: A novel dataset for personalized image aesthetic assessment

## Abstract

*We present the Large Art Personalized Image Set (LAPIS), a novel dataset for personalized image aesthetic assessment (PIAA). It is the first dataset with images of artworks that is suitable for PIAA. LAPIS consists of 11,723 images and was meticulously curated in collaboration with art historians. Each image has an aesthetics score and a set of image attributes known to relate to aesthetic appreciation. Besides rich image attributes, LAPIS offers rich personal attributes of each annotator. We implemented two existing state-of-the-art PIAA models and assessed their performance on LAPIS. We assess the contribution of personal attributes and image attributes through ablation studies and find that performance deteriorates when certain personal and image attributes are removed. An analysis of failure cases reveals that both existing models make similar incorrect predictions, highlighting the need for improvements in artistic image aesthetic assessment.*

## 1. Introduction

Computational aesthetics is a subfield of computer science that focuses on the automated aesthetic assessment of images [16]. The current trend is to leverage deep learning to perform image aesthetic assessment (IAA). Although several IAA datasets [7, 9, 10, 21, 35, 37] were created in the last decade, existing datasets often come with limitations. Many of these datasets were created by scraping photography/art contest websites [15, 23, 37]. The aesthetic annotation is then derived from the number of likes or votes an image receives. This may introduce biases in the data, for example: (1) the images in these datasets are all highly aesthetic because unaesthetic images will rarely be submitted to a contest, (2) the votes may be influenced by the amount of engagement (*e.g.* number of views or downloads). Those who vote may simply not see images that may be equally or more aesthetic. As a result, the aesthetic annotations may not span the entire spectrum of aesthetics and may not represent aesthetic appreciation accurately.

Another limitation of many existing datasets is that they

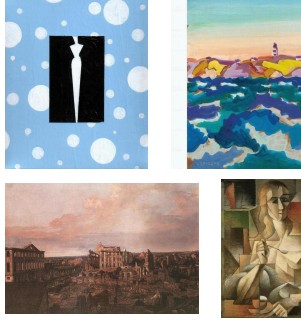
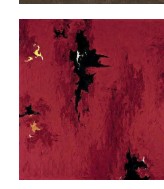
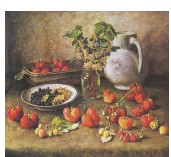

Figure 1. Illustration of the image selection. Images on the left were excluded during the quality check. The top left image contains a watermark and the bottom left image is a sculpture. The images on the right are example images in LAPIS.

average out individual differences [7, 10, 15, 37]. Aesthetic assessment is a rather subjective task, rendering it difficult to model and predict. Many existing datasets treat the individual variation as noise and compute an average aesthetic score for a given image. Predicting these average aesthetic scores using machine learning is referred to as generic image aesthetic assessment (GIAA). These datasets can advance research to understand universal properties that drive aesthetic appreciation. However, given the subjective nature of aesthetic appreciation, personalized image aesthetic assessment (PIAA) may offer a more encompassing framework.

PIAA concerns the prediction of aesthetic scores for each annotator separately [21]. This is a very useful task from a marketing perspective, with applications like personalizing advertisements based on individuals' online presence (*e.g.* likes on social media). However, the current PIAA datasets all consist of natural images.

Art has been largely under-explored in computational

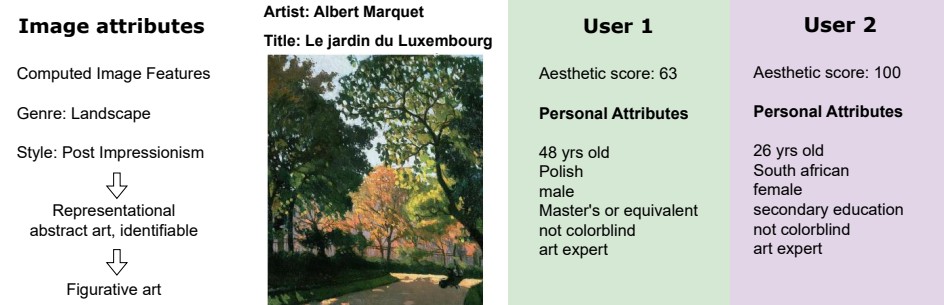

Figure 2. Visualisation of the types of data in LAPIS. All images have metadata (title, artist) and image attributes. Images are rated by multiple users on their aesthetic appeal. For each user, we have a set of personal attributes

aesthetics [37]. In fact, none of the existing PIAA datasets include artistic images. Yet, previous research found that individual differences in aesthetic appreciation are larger for artistic images than photographs [29, 30]. Therefore, an artistic dataset is more suited to tackle the problem of PIAA. In addition, artistic image aesthetic assessment (AIAA) presents a relevant challenge for computer vision due to the complexity of artistic images and their need for better pre-processing methods [27]. AIAA has relevant applications given the increase in online art trading [14] and user-friendly technology such as DALL-E [3] which allows almost anyone to generate artistic images. Our contributions are as follows:

- We present the first artistic dataset for PIAA, called the Large Art Personalized Image Set (LAPIS). Each image in LAPIS was rated by on average 24 annotators and includes rich personal and image attributes to inform and improve personalized predictions.

- Our dataset establishes a new standard for data quality in the field. LAPIS was meticulously curated in collaboration with art historians and addresses the limitations mentioned above that are present in many of the existing datasets.

- We analyze the data and perform experiments for both GIAA and PIAA. We find that our data quality improves GIAA and training with rich image and personal attributes improves PIAA.

## 2. Related work

### 2.1. Art datasets

There are a few well-curated datasets with art images and aesthetic annotations from the field of empirical aesthetics (VAPS [5], JenAesthetics [2]). Unfortunately, the number of images in these datasets is relatively small (999 and 1628 respectively), rendering them insufficient for machine learning applications. On the other end of the spectrum

are the large artistic datasets without aesthetic annotation [1, 22, 33]. More recently, datasets designed for IAA started to include more artistic images [7, 37]. The BoldBrush Artistic Image Dataset (BAID) [37] is the largest collection of artistic images with aesthetic annotations. It consists of over 60K images of artworks. Images are sourced from the website "BoldBrush"[1], a platform that allows artists to share their work online. BoldBrush hosts monthly art contests, where users can vote for the artistic images they like. The images in BAID received 360,000 votes in total. These votes were then transferred into a score representing aesthetics, where a higher number of votes translates into a higher aesthetic score. As such, BAID offers a large dataset for GIAA. However, it is not suitable for PIAA, given that scores are obtained by counting votes. Additionally, these votes may not represent aesthetics accurately, highlighting the need for large, well-curated datasets that contain artistic images.

### 2.2. Datasets for personalized image aesthetic assessment (PIAA)

Datasets for PIAA include a user ID which allows tracking of responses of a single annotator across different images. The FLICKR-AES [21] dataset was the first dataset introduced for PIAA and consists of 40K images which are scored by at least 5 annotators each. The images in the dataset are photographs sourced from the photography website FLICKR[2]. More recently, the Pairwise-Relabeled Aesthetic Attribute Dataset (PR-AADB) [6] was introduced as a test set for PIAA. It is a relabeled version of the AADB dataset [10] which is used for GIAA and contains rich image attributes. 165 annotators judged the images in a pairwise preference task, resulting in 16k labeled image pairs. The dataset was created to test for robustness in PIAA and can be used for few-shot personalization.

The Explainable Visual Aesthetics dataset (EVA) [9]

---

[1]https://faso.com/boldbrush/popular
[2]https://www.flickr.com/

| Figurative *(7976)* | | Abstract *(3747)* | |
|---|---|---|---|
| Representational figurative art *(5131)* | Representational abstract art - identifiable *(2845)* | Non-representational abstract art - lyrical *(3202)* | Non-representational abstract art - geometric *(545)* |
| Early Renaissance *(91)*
High Renaissance *(146)*
Northern Renaissance *(557)*
Mannerism (Late Renaissance) *(212)*
Baroque *(409)*
Rococo *(435)*
Romanticism *(438)*
Realism *(531)*
Art Nouveau *(412)*
Symbolism *(489)*
Pop Art *(357)*
New Realism *(152)*
Contemporary Realism *(301)*
Naïve Art / Primitivism *(602)* | Impressionism *(499)*
Post-Impressionism *(418)*
Pointillism *(281)*
Fauvism *(442)*
Cubism *(427)*
Synthetic Cubism *(203)*
Analytical Cubism *(79)* | Abstract Expressionism *(1839)*
Action Painting *(92)*
Color Field Painting *(1274)* | Minimalism *(541)* |

Table 1. The styles represented in LAPIS at different granularity levels. The lowest level includes the 27 styles that were originally in WikiArt. The overarching styles were defined by art historians to indicate the level of abstractness of the styles for a non-expert audience. The number of images per style is indicated after each style label.

provides both image attributes and personal attributes. Although EVA is not typically used for PIAA, it does include demographic information about the annotators that allows for PIAA. It consists of 40K photographs with an average of 30 annotators per image. The images were rated on various relevant attributes for aesthetics, alongside aesthetic appreciation itself. Participants were asked to indicate how much they liked the following attributes: light and color, composition and depth, quality and semantics. Annotators were then asked to indicate for each image how much their aesthetic rating was influenced by each of the attributes. In terms of personal attributes, the dataset includes the age, gender, region and photographic level of the annotators.

The PARA [35] dataset similarly offers both image attributes and personal attributes. It consists of 30,000 images annotated by 438 subjects with an average of 25 annotators per image. The images were sourced from Flickr and Unsplash[3], as well as existing datasets with aesthetic annotations. These existing aesthetic annotations were used to sample images from all aesthetic levels to balance the aesthetic score distribution. They used automated scene classification to balance the images across content. Images were annotated on aesthetic appeal, quality and a set of image attributes (color, composition, depth of field, content, light, object emphasis). They additionally collected emotion attributes and content preferences, as well as demographic information about the annotators. The demographic information includes age, gender, education level, artistic and photographic experience and scores on the Big Five personality test [19]. As such, the PARA dataset is the first to offer rich attributes, both at the image level and the personal level.

## 2.3. Personalized image aesthetic assessment (PIAA)

Many research efforts in PIAA have been focused on predicting an aesthetic score per annotator (usually referred to as 'user' in the context of PIAA) without informing this decision by personal attributes such as demographics [13, 18, 21, 31, 32, 36]. Rather, many works rely on image attributes to improve personalized predictions. One of the earliest works by Ren *et al*. [21] included image attributes to inform personalized aesthetic predictions. They created the FLICKR-AES dataset which had ratings of 5 different individuals for each image. In their pipeline, they predicted image attributes as well as a generic aesthetic score. These attribute predictions were then used to predict an offset from the generic aesthetic score, to obtain a personalized score for each of those 5 individuals. In a similar vein, more recent work [34, 41] leveraged image attributes to improve predictions in PIAA. Li *et al*. [11] shifted from this focus on image attributes to personality traits that may influence aesthetic assessment. They trained a siamese network to jointly learn generic aesthetic scores and personality traits. These were then fused to predict a personalized aesthetic score given a personality trait. Zhu *et al*. [39] similarly leveraged personality prediction to improve PIAA. Their model is informed by both image attributes and personal attributes.

Hou *et al*. [8] and Zhu *et al*. [40] extended this idea, by modeling *interactions* between image features and personal attributes. Hou *et al*. [8] used an interaction matrix in their pipeline to model interactions between image features and individual raters' preferences for these image features. Zhu *et al*. [40] consider interactions between demographic traits and learned image attributes. Their model, referred to as PIAA-MIR, is trained on the PARA dataset which is the

---

[3]https://unsplash.com/

| image dimensions | complexity/lightness/contrast | color | symmetry/balance | fractality/self-similarity | entropy/feature distribution |
|---|---|---|---|---|---|
| image size | RMS contrast | color entropy | **pixel-based:** | **Fourier spectrum:** | anisotropy |
| aspect ratio | lightness entropy | **channel means:** | mirror symmetry | slope | homogeneity |
|  | complexity | RGB | DCM | sigma | **edge-orientation entropy:** |
|  | edge density | lab | balance | **fractal dimension:** | 1st order |
|  |  | HSV | **CNN-feature-based:** | 2-dimensional | 2nd order |
|  |  | **channel standard deviation:** | left-right | 3-dimensional | **CNN feature variance:** |
|  |  | RGB | up-down | **self-similarity:** | sparseness |
|  |  | lab | left-right AND up-down | PHOG-based | variability |
|  |  | HSV |  | CNN-based |  |

Table 2. Overview of the image attributes available in LAPIS, computed with the toolbox by Redies *et al*. [20]

only dataset rich in both image attributes and personal attributes.

Lastly, Shi *et al*. [24] extended this idea by considering interactions both within and between these two types of attributes (personal and image attributes). They used graph neural networks to perform collaborative filtering on the PARA dataset. Their model is referred to as PIAA-ICI and achieves state-of-the-art performance, together with the model by Zhu *et al*. [40]. They are the only two models (to the best of our knowledge) that inform PIAA with rich personal and image attributes. Therefore, we implemented these two models to perform experiments on LAPIS (see section 5).

## 3. Methods

### 3.1. Image selection

Images were sourced from WikiArt[4], an online archive of artworks that is constructed with the aid of galleries or museums. Similarly as the better-known Wikipedia, gallery or museum curators could contribute to the archive by uploading images of their artworks alongside metadata. LAPIS is a selection of 11,723 images from the WikiArt paintings dataset, which comprises mostly paintings but additionally includes some sketches. LAPIS includes 26 styles (ranging from Renaissance to Minimalism) and 7 genres (abstract, cityscape, flower painting, landscape, nude painting, portrait and still life). We selected images from those 7 genres, since they correspond well to the content that is displayed (as opposed to the remaining genres 'religious painting', 'genre painting' and 'sketch and study'). We added hierarchical style labels informed by art historians to provide clarity regarding which styles are closer in terms of abstractness (see Table 1). Given the interdisciplinary nature of computational aesthetics, these labels provide contextual information for those without a background in art history.

When selecting images, we prioritized those with a higher resolution and a more balanced aspect ratio. The final selection is (largely) balanced[5] for genre when portrait is combined with nude painting and flower painting is

combined with still life. There are a larger number of figurative works (7976) as opposed to abstract works (3747) in LAPIS, as we tried to sample a representative number of works from each style with regards to the total number of works in the full WikiArt dataset.

As a quality check of the data, we manually checked each image in a first small selection of 1990 images. We saw that the dataset included some provocative images, sculptures, duplicates and images containing text (*e.g.*, from a watermark or copyright mark, see Figure 1). We manually removed these instances. Some images included the frame around the artwork, which we cropped manually. We noticed that the genre did not always describe the content of the image correctly. We added a content label and manually described what was most salient in the image (corresponding to one of the 7 genre categories). In addition, we noticed that some of the style labels in WikiArt were inaccurate. We manually adjusted them with the assistance of art historians in this smaller set. Based on this check, we automated the removal of duplicate images, frames of artworks and images containing text in the remainder of our image set (details can be found in the supplementary material). We manually checked the images in the style categories 'abstract expressionism' and 'minimalism' since these had the highest number of sculptures in our smaller sample. We removed every instance that was not a painting or sketch in these two style categories. We had noticed that most of the inaccuracies in genre were the 'flower painting' label being used for other genres. Therefore, we manually checked all the images labeled as 'flower painting' in the larger set and corrected the style label if needed.

### 3.2. Online study

We set up an online study to obtain aesthetic evaluations for the images in LAPIS. We recruited 552 participants through Prolific[6], a UK based platform allowing workers to anonymously participate in online studies. Prolific is known for having more reliable workers and more safeguards against bots, as well as providing fair payments to its workers. We obtained ethical approval for the study. Only those with achromatopsia (a condition that affects one's ability to per-

---

[4]https://www.wikiart.org/

[5]Further details regarding the distribution of LAPIS can be found in the supplementary material.

[6]https://www.prolific.com/

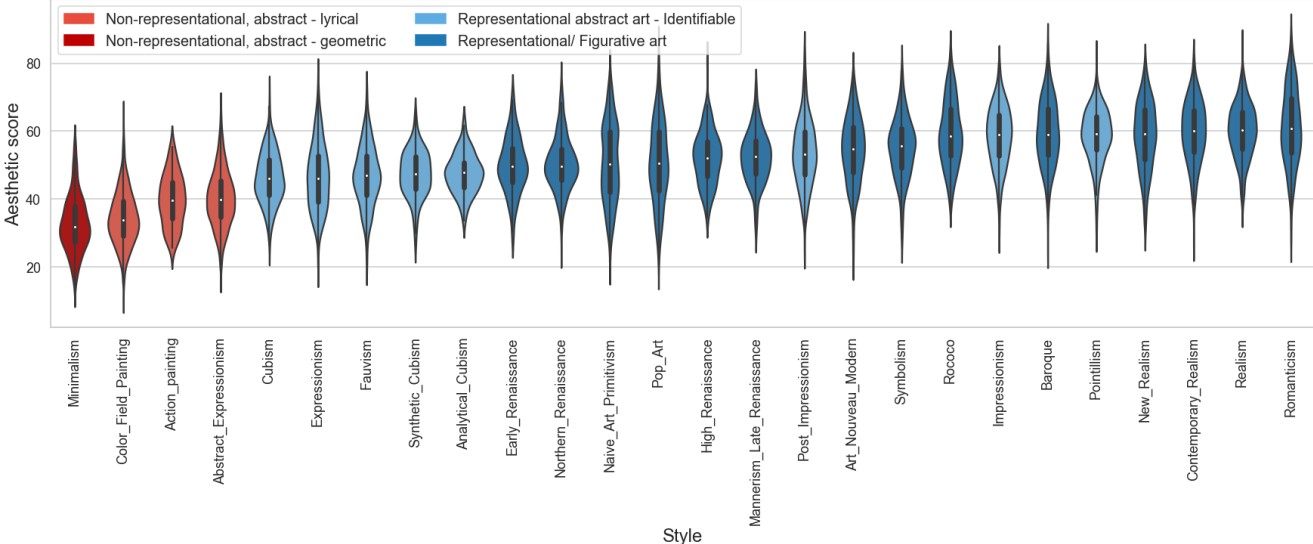

Figure 3. Violin plots of the data distribution per style. Violins are ordered from lowest median (top) to highest median aesthetic scores (bottom). The abstract and figurative styles are shown in different shades of red and blue respectively.

ceive colors) were excluded from participation. At the start of the study, participants provided their informed consent and answered demographic questions (see section 3.3). A set of example images were shown to indicate what kind of images to expect during the study. There was a practice trial before the actual trials in which participants rated the aesthetic value of the displayed images. After rating a block of images, participants were asked to indicate how many images they recognized. After removal of non-conscientious participants, the average number of annotators per image was 24. Further details regarding the annotation procedure can be found in the supplementary material.

### 3.3. Attributes

Figure 2 shows an example image in LAPIS with all its metadata and attributes. LAPIS includes both personal and image attributes. In terms of personal attributes, each annotator was assigned an ID and provided their age, nationality, gender and education level. We asked whether they are colorblind and measured their art interest using the art interest subscale of the VAIAK [25, 26]. Art familiarity was assessed by asking participants how many images they recognized after each block of approximately 250 images. Annotators were divided into art experts and art novices based on their art interest and art familiarity (see section 4.3)

The image attributes include metadata (style and genre) and computed image attributes. We used the toolbox by Redies *et al*. [20] to compute these attributes. It computes 31 image attributes that are known to matter for aesthetic appreciation. Table 2 gives an overview of the image attributes, ordered as in Redies *et al*. [20]. The attributes re-

late to the image dimensions, complexity, balance, color, luminance, contrast, lightness, symmetry, fractality, self-similarity, entropy and feature distribution. Some of the attributes are related to multiple computed image *features*. For example, the color channel means for the RGB color spectrum computes 3 values, *i.e.* one mean value for each channel. As such, there are 47 image features per image, relating to 31 image attributes. For more detailed information on specific features and their relevance for aesthetics, we refer the reader to the original work by Redies *et al*. [20].

## 4. Analysis of LAPIS

### 4.1. Personal attributes

We found a moderate correlation between aesthetic score and art interest ($r = 0.35, p < 0.01$). Figure 4 shows the mean aesthetic rating given by a participant in function of their art interest score. Participants who scored higher on art interest rated the images higher on average. None of the other personal attributes revealed strong differences in aesthetic scores.

### 4.2. Image attributes

Figure 5 displays the histograms of aesthetic scores for figurative and abstract works where scores are averaged per image (as in GIAA). The data seem normally distributed, with more images receiving a mean rating around the middle of the rating scale. This is a similar trend as in most IAA datasets, and is partially due to people's tendency to avoid the extremes of rating scales [4] and set effects [12, 17, 28]. We also see a clear trend of preferences towards more figu-

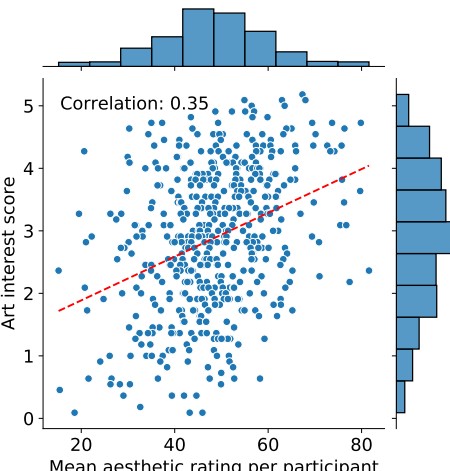

Figure 4. Scatter plot of the mean aesthetic rating given by an annotator in function of their art interest score (as measured with the VAIAK familiarity questionnaire [25, 26]). The marginal distributions for both art interest and aesthetic scores are shown on the side. We found a correlation between art interest and aesthetic scores ($r = 0.35, p < 0.01$).

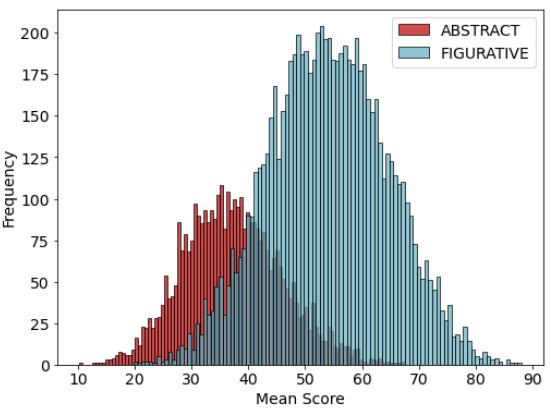

Figure 5. Histogram of the aesthetic scores averaged per image. Data corresponding to abstract and figurative works is shown in red and blue respectively. We observe a trend towards preferences for figurative works.

rative works. This is further highlighted in Figure 3, showing the score distribution for each style. The styles are ordered from lowest median score to highest median score. We observe that the four abstract styles received the lowest median scores, whereas the highest scoring styles are among the most figurative styles (e.g. Realism). To assess the robustness of this trend, we looked at agreement between annotators per image. Figure 6 shows the distribution of standard deviations in scores per image in function of the mean score of that image. In general, we can see that images with a mean score that is at the end of the rating

scale (either highly aesthetic or unaesthetic) tend to have lower standard deviations, meaning raters agree more on their evaluation of these images (in line with previous work [15]). Strikingly, all the images with a low average score are abstract works, whereas all the images with high average scores are figurative works. There is a small trend towards higher standard deviations for abstract works, meaning annotators disagreed more when judging those works. We saw a similar trend of preferences for certain genres. Abstract works were judged more negatively, while landscapes and cityscapes tend to receive higher ratings (Figure s18). Lastly, we found that luminance entropy and edge orientation entropy correlate positively with aesthetic scores ($r = 0.47; r = 0.45, p < 0.01$), while sparseness and CNN symmetry correlate negatively with aesthetic scores ($r = -0.40; r = -0.48, p < 0.01$) (Figure s19). This suggests that annotators preferred more complex works with higher levels of entropy and less symmetry over more simple works. In terms of color, we found that color channel means tend to correlate negatively with aesthetic scores while color channel standard deviations correlate positively with aesthetic scores. This indicates that annotators rated colorful works higher than those with more uniform colors.

### 4.3. Personal x Image attributes

We looked at possible interactions between personal and image attributes. Art interest was the only personal attribute that correlated with aesthetic scores. We found that none of the computed image attributes correlated with art interest. When looking at image style and genre, we found that art interest relates to the difference in aesthetic scores for abstract works (Figure s17). We divided the data into a group of novices and experts using a median split based on their scores on art interest as primary variable and the amount of images they recognized as secondary variable. We observe that novices tend to score abstract works consistently lower, whereas this is less apparent for experts.

### 5. Experiments

### 5.1. GIAA

We divided LAPIS into a train, validation and test set using a 70/10/20 split. We used stratified sampling based on aesthetic score and style to ensure that the test set is representative of the training set. Both the test and validation set resemble the distribution of the training set well in terms of aesthetic score, style and genre (more details can be found in the supplementary material).

A representative test set is important to accurately assess a model's performance. Given that the data is normally distributed, a model that predicts scores around the mean would still achieve decent performance. As a result, a test set that is not representative may lead to misleading

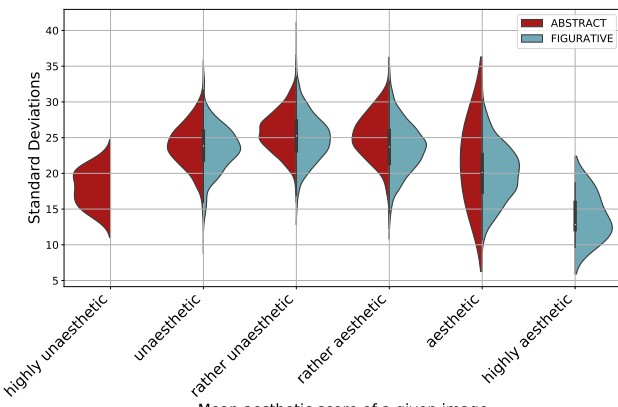

Figure 6. The distributions of standard deviations per image across the rating scale. Results are shown in red for abstract works and in blue for figurative works.

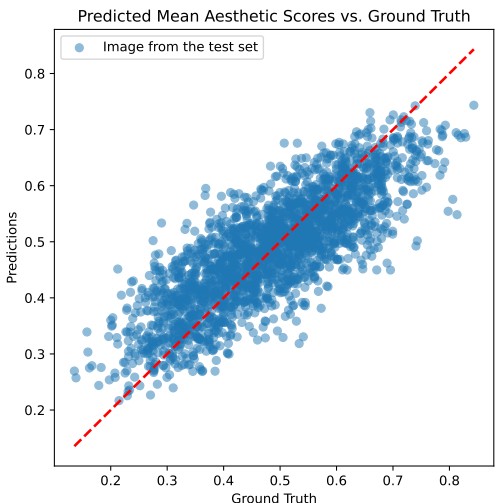

Figure 7. Test set predictions of ResNet50 trained on LAPIS.

| PARA | single user evaluation scheme | |
|---|---|---|
| PIAA-MIR | $0.716 \pm 0.0008$ | |
| PIAA-ICI | $0.739 \pm 0.0011$ | |
| LAPIS | traditional train/test split | 4-fold cross-validation |
| PIAA-MIR | 0.6958 | $0.2793 \pm .0215$ |
| PIAA-ICI | 0.6941 | $0.2773 \pm .0235$ |

Table 3. Comparison of the state-of-the-art models on LAPIS vs PARA. The top rows are the results reported in [24, 40]. The bottom rows are the results on LAPIS. The left column are results obtained by using a traditional evaluation scheme with a train, validation and test split of the images. The right column reports the results of a 4-fold cross-validation scheme were there is no overlap of users in test and train data.

works), we predict a score for a given combination of demographics. The aim of this evaluation scheme is to assess the models' ability to inform predictions by a set of personal and image attributes only, without knowledge of other scores given by the same annotator. The goal is to create models that make more general predictions and predict scores for unseen users better. Similarly as in GIAA, we divide the data into train, validation and test sets. We train on the full training set without training per annotator, implying that there is overlap in annotators between the train and test set. When evaluating the models, we assess performance on the full test set. One could argue that this is an unfair evaluation, given that the model is not tested on a set of unseen users (solely unseen images). Therefore, we consider an alternative evaluation scheme where we introduce 4-fold cross-validation to select separate train and test annotators. Specifically, the train set consists of (training images, train users), the validation set of (validation images, train users), and the test set of (test images, test users). Table 3 shows the results. The results using our naive evaluation scheme are close to results obtained on the PARA dataset in the original work by Zhu *et al*. [38] and Shi *et al*. [24]. This minor difference in performance may relate to the fact that art images are more challenging for PIAA due to the higher subjectivity of the ratings [29, 30]. When we use the 4-fold cross-validation, performance drops significantly. This suggests that the model overfits on training users using the naive evaluation scheme. It highlights the need for better methods to create models that generalize well to unseen data.

### 5.3. Ablation of attributes

To further understand how the image attributes and personal attributes contribute to the predictions, we perform ablation studies by removing an attribute as input during training. The results are shown in Table 4. In terms of personal attributes, we performed the ablation study only with art interest and age, given that these were the personal attributes

interpretations of the performance metrics. By using stratified sampling, we ensure that the test set in LAPIS spans the entire range of scores and does not contain an over-representation of styles that may be easier to predict. Figure 7 shows predictions on the test set of ResNet50 trained on LAPIS. We can see that the model predicts scores well across the full range of possible scores.

### 5.2. PIAA

We implemented both PIAA-MIR [38] and PIAA-ICI [24] and trained them on LAPIS. Our implementation is as close to the original work as possible, however, the personal and image attributes are replaced with the attributes in LAPIS. Rather than evaluating a single user (as in all previous PIAA

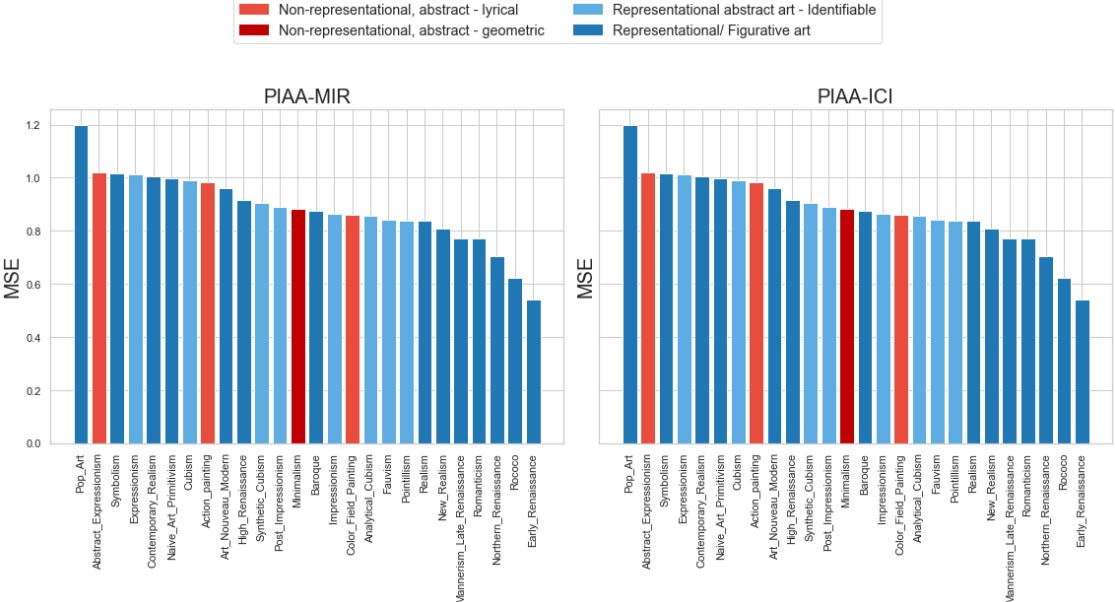

Figure 8. Barplot displaying the mean MSE per style for PIAA-MIR [40] and PIAA-ICI [24] trained on LAPIS. The styles are color coded based on our style division of four styles ranging from fully figurative (blue) to fully abstract (red).

that correlated the most with aesthetic scores in LAPIS. We observe that the omission of art interest deteriorates performance, indicating that this attribute informs the predictions of the model. We do not see such an effect for age. In terms of image attributes, we observe that the omission of style and genre labels deteriorates performance, indicating their importance for aesthetic evaluation. Interestingly, we do not see a decrease in performance when the objective image features that are known to relate to aesthetics are removed as inputs. We hypothesize that this may be due to the backbone already extracting these features (or correlated features) in its convolutional layers.

### 5.4. Analysis of failure cases

Lastly, we checked for which image and personal attributes the models struggle to predict aesthetic scores accurately. Figure 8 shows the mean MSE of the images in the test set per style. Although the challenging styles include both figurative and abstract styles, the top-5 best-predicted styles are all representational figurative art. Prediction errors are higher for disliked genres and lower for liked genres (Table s5). In terms of personal attributes, we do not find a correlation between the MSE of predictions and art interest. We do, however, find a negative correlation between prediction errors and age ($r = -0.33, p < 0.01$ for PIAA-ICI and $r = -0.40, p < 0.01$ for PIAA-MIR), indicating that the models make more prediction errors for older users. This can be in part explained by the over-representation of younger annotators in LAPIS.

| Ablation | SROCC |
|---|---|
| Baseline | 0.69583 |
| Art interest | **0.55155** |
| Age | 0.68978 |
| Style and genre | **0.55851** |
| Objective image attributes | 0.70118 |

Table 4. Results of our ablation studies. The left column indicates which attribute is removed. The right column shows the SROCC for the given ablation. We observe that performance deteriorates when we remove art interest of the personal attributes and style and genre of the image attributes.

## 6. Conclusion

We present a novel dataset with artistic images for PIAA, which is the first of its kind. We created LAPIS with art images which is more suited for PIAA given the larger individual differences in the assessment of artistic images. LAPIS is well-curated and contains rich personal and image attributes. We show that the high-quality data in LAPIS result in good performance on GIAA using a simple resnet50. PIAA presents a much more challenging task. Our experiments show that the inclusion of rich personal and image attributes improve predictions in PIAA. However, we find that existing models fail on unseen users and images, indicating that PIAA remains a challenging task.

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
