# LAPIS: A novel dataset for personalized image aesthetic assessment

## Supplementary Material

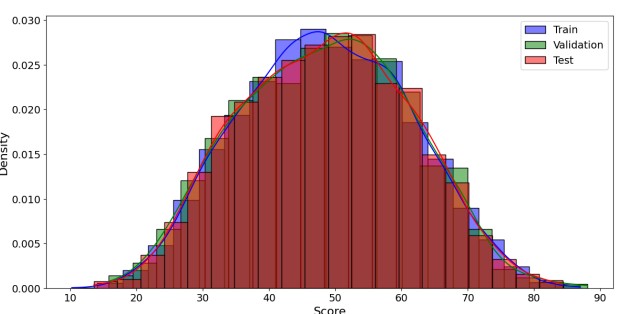

Figure 1. Distributions of aesthetic scores in LAPIS. The distribution of each data partition (train, validation and test) is shown in different colors.

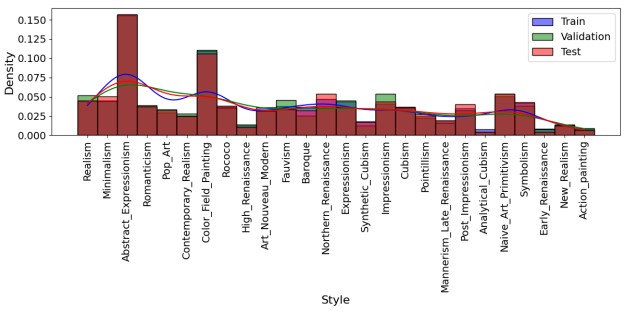

Figure 2. Distributions of styles in LAPIS. The distribution of each data partition (train, validation and test) is shown in different colors.

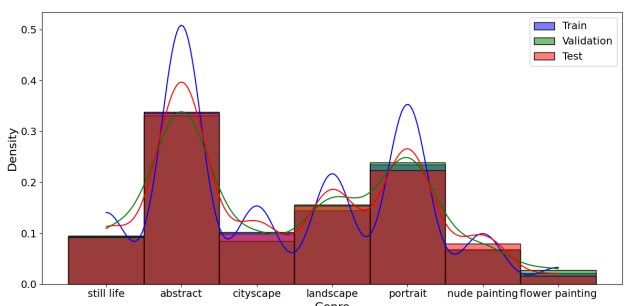

Figure 3. Distributions of genres in LAPIS. The distribution of each data partition (train, validation and test) is shown in different colors.

## 1. Data distribution LAPIS

Figure 1 shows the distributions of aesthetic scores for each data partition (train, validation and test set). We used stratified sampling based on aesthetic score and style (at the superordinate level, i.e. figurative vs abstract) to ensure that the test set is representative of the training set. Figure 2 and Figure 3 show the distribution of styles and genres, respectively, in LAPIS per data partition. We observe that the test set resembles the distribution of the training and validation set well for aesthetic score, style and genre.

## 2. Quality checks

### 2.1. Automating image curation

We automated the process of removing frames in the larger image set of LAPIS. We applied code by Robert A. Gonsalves on github[1] that was created to remove frames of paintings in the WikiArt dataset.

Detecting duplicate images was done using the difPy package. We removed 21 duplicate images.

Text detection in the images was done using pytesseract [2]. 1,927 images were flagged by pytesseract and were subsequently removed from our set.

## 3. Online study procedure

The study was programmed using the JsPsych library [1] in Javascript. At the start of the study, participants provided their informed consent for the study. They were asked if they have a form of colorblindness or have normal eyesight. Only those with achromatopsia (a condition that affects one's ability to perceive colors) were excluded from participation. They answered a set of demographic questions and completed the art interest part of the VAIAK questionnaire [3, 4] (see section 4.1). A set of example images were shown to indicate what kind of images to expect during the study. There was a practice trial before the actual trials in which participants rated the aesthetic value of the displayed image using a visual analogue scale with 7 tick points (see Figure 4). After rating a block of images, which consisted on average of 250 images and took around 30 minutes, participants were asked to indicate how many images they recognized. In a first wave of data collection, participants could choose to stop the study after one block or continue rating images (up to a maximum of 8 blocks). Because this complicated the payments on Prolific, the second wave of data collection consisted of exactly 2 blocks for every participant which took on average 1 hour to complete.

---

[1]https://github.com/robgon-art/MachineRay

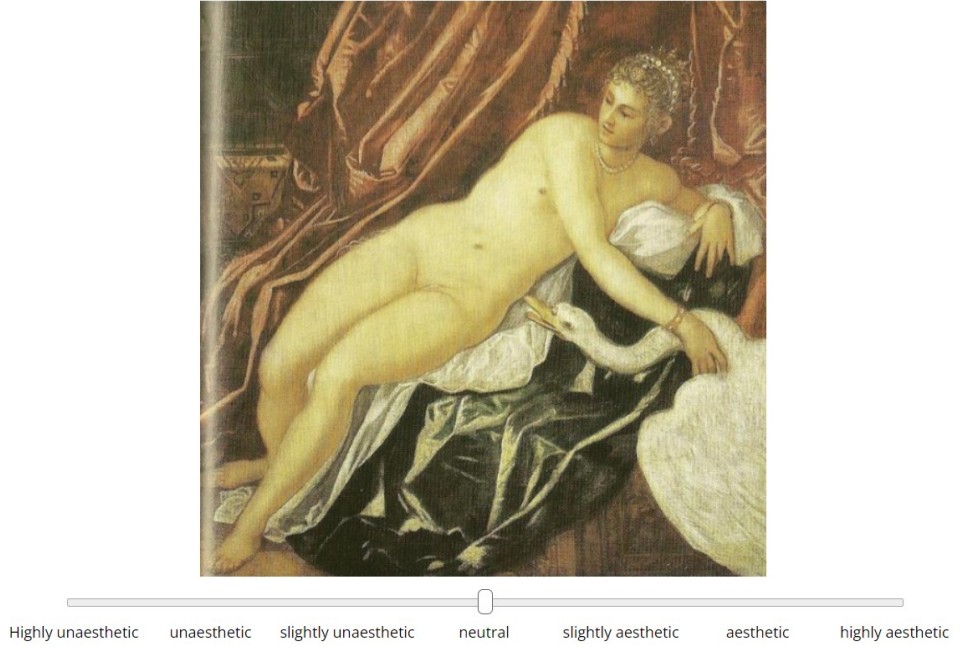

Figure 4. An example trial in the online study.

## 3.1. Cleaning the data

Since there is no right or wrong answer when it comes to aesthetic appreciation, it was not trivial to determine exclusion criteria to detect non-conscientious trials. One could argue that the size of the dataset is sufficiently large to provide reliable trends in group differences regardless of the noise introduced by such non-conscientious trials. Therefore, we used rather lenient criteria to exclude only those trials who are almost certainly non-conscientious. Participants who gave the same response (i.e. a specific value on the scale that was turned into integers from 0 to 100) more than 100 times were flagged. Those who gave the same rating over 50% of the experiment, suggesting participants were not rating aesthetic value conscientiously, were removed entirely. When participants gave the same response 15 times in a row (or more), those trials were removed. This led us to exclude five participants based on the first criterion, which amounted to the removal of 2160 trials. None of the remaining participants met the second criterion.

## 4. Personal attributes

### 4.1. Study Procedure

At the beginning of our online study, participants were asked a set of demographic questions. Participants could indicate their age and nationality from a list of all sensible options (*e.g.* 0-100 for age). The response options for gender were "female", "male", "non-binary", "other/would prefer not to disclose". The response options for the level of education were "primary education", "secondary education", "bachelor's or equivalent", "master's or equivalent" and "doctorate". Participants were additionally asked to indicate whether they are colorblind with response options "no", "yes, but I still perceive colors" or "yes, and I do not perceive any colors". Since those with achromatopsia were discouraged to participate in the study, none of our participants indicated that they are fully colorblind. Out of the annotators in LAPIS, 1.2% is colorblind but still perceives colors. After rating a block of approximately 250 images, participants were asked to indicate how many im-

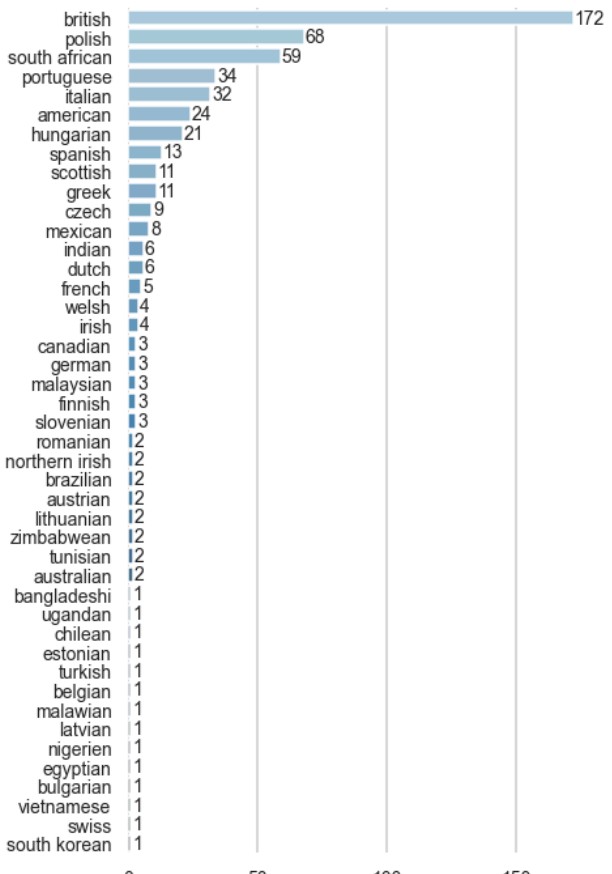

Figure 5. Histogram of the nationalities of annotators in LAPIS.

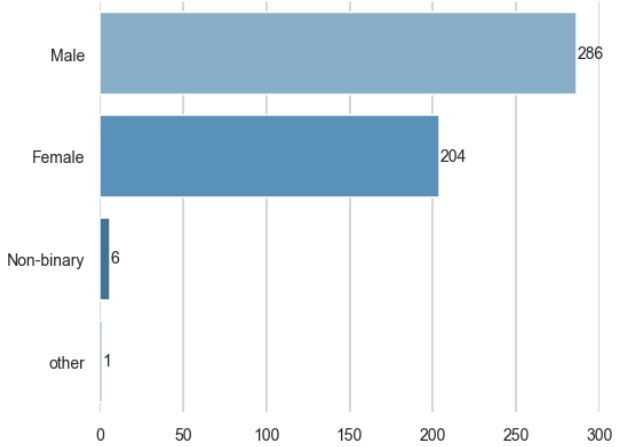

Figure 6. Histogram of the genders of annotators in LAPIS.

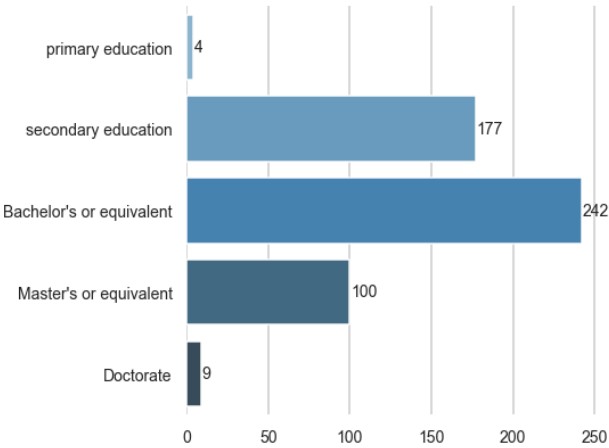

Figure 7. Histogram of the education levels of annotators in LAPIS.

ages they recognized. The response options were "none", "1-10", "11-25" or "more than 25".

## 4.2. Descriptive statistics

Figure 6 shows the gender occurrences of the annotators[2] in LAPIS. Although the data are relatively balanced between male and female annotators, nonbinary individuals are underrepresented in LAPIS. Figure 8 shows the ages of annotators. Our data includes mostly younger individuals. Figure 5 shows the nationalities of the annotators. The large number of British annotators can be in part explained by the fact that we ran the study on Prolific, which is a UK based platform. Lastly, Figure 7 shows the education level of the annotators, which seem to be representative for the larger population.

---

[2]It should be noted we do not have all the demographic information for all annotators in the dataset. Therefore, the occurrences in these plots do not sum up to the same number of annotators for all plots.

## 5. Analysis of LAPIS

We find a general trend of lower aesthetic scores for abstract works. Figure 9 shows that abstract works score lower than figurative works, and this trend is stronger for novice annotators. Figure 10 shows a similar trend for the different genres, with abstract works scoring the lowest compared to landscapes and cityscapes.

Figure 11 shows the correlations between aesthetic scores and computed image attributes. Attributes are ordered from highest to lowest Pearson correlation coefficient. The highest correlating attributes are luminance entropy and edge-orientation entropy, suggesting a preference for works with rich textures or complex compositions. Sparseness and CNN symmetry (up-down) correlate negatively with aesthetic score, suggesting that annotators disliked simple and symmetric works.

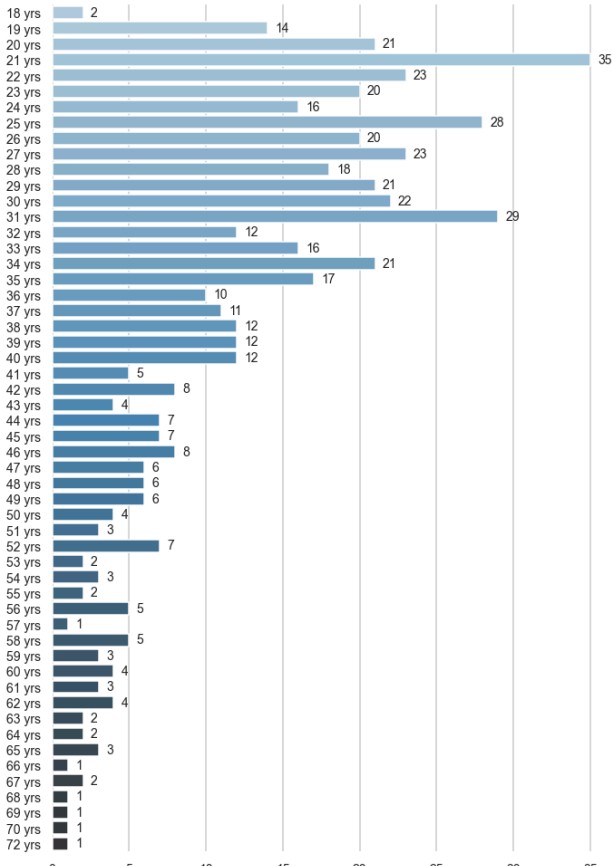

Figure 8. Histogram of the ages of annotators in LAPIS.

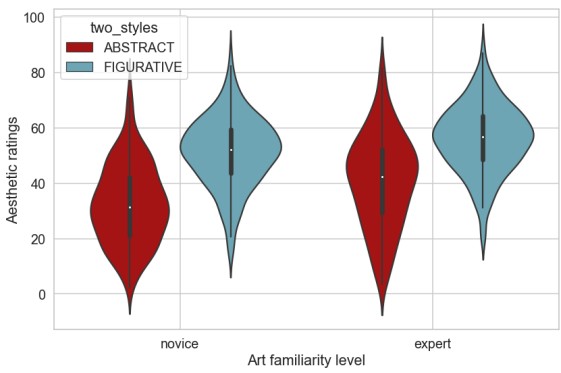

Figure 9. Violinplot comparing the mean ratings given by novices and experts for figurative and abstract works.

## 6. Failure cases

Table 1 shows the mean MSE per genre on LAPIS' test set. We observe that the three most disliked genres result in a higher MSE, whereas, the four most liked genres result in lower MSE scores for both PIAA-MIR and PIAA-ICI.

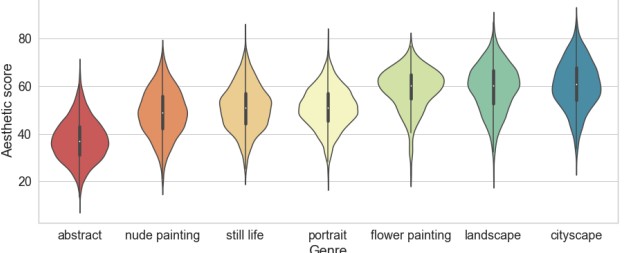

Figure 10. Violin plots of the data distribution per genre. Violins are ordered from lowest median to highest median aesthetic scores.

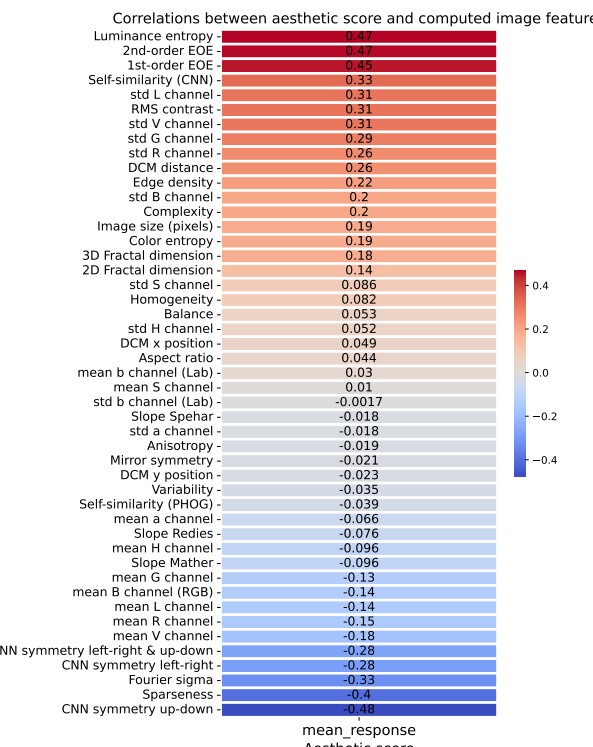

Figure 11. Pearson correlation coefficients between aesthetic scores and computed image attributes.

| Genre | PIAA-MIR | PIAA-ICI |
|---|---|---|
| Nude painting | 1.01292 | 1.02533 |
| Still life | 0.96589 | 0.99588 |
| Abstract | 0.93523 | 0.94184 |
| Landscape | 0.87165 | 0.86954 |
| Cityscape | 0.84947 | 0.87059 |
| Portrait | 0.81820 | 0.82269 |
| Flower painting | 0.73471 | 0.73106 |

Table 1. Mean MSE on LAPIS' test set per genre for both PIAA-MIR and PIAA-ICI.