# OpenReview forum: "LAPIS: A novel dataset for personalized image aesthetic assessment"
_thecvf.com/CVPR/2025/Workshop/CVEU — CVPR 2025_

### Official Review · Reviewer_2yN1 · 2025-03-13
**Interesting unconventional dataset, why not?**

**Rating:** 4
**Confidence:** 4

**Review:**

It's NOT about video creation, editing or understanding. But, it's about topics of interest for some members of our community: art, and image aesthetics. Its novelty in addressing personalized aesthetic assessment with a rich, well-curated dataset fills an important gap in current research. The experiments and insightful ablation studies further bolster its impact. While there are areas for improvement—particularly regarding the clarity of dataset curation, and discussions on bias—the overall contributions are significant and promising.

**Executive Summary.** This submission introduces LAPIS, a novel dataset specifically designed for personalized image aesthetic assessment (PIAA) in the artistic domain. Unlike existing IAA datasets that predominantly feature natural images, LAPIS comprises 11,723 artistic images curated from WikiArt and enriched with comprehensive metadata. Each image is annotated not only with detailed image attributes—such as style, genre, and a suite of computed features—but also with rich personal attributes of the annotators (e.g., demographics, art interest, and familiarity). The dataset was curated in collaboration with art historians to ensure high data quality and relevance. The authors conduct extensive experiments, implementing both Generic Image Aesthetic Assessment (GIAA) using a ResNet50 model and state-of-the-art PIAA models (PIAA-MIR and PIAA-ICI). Ablation studies further reveal that personal attributes, particularly art interest, and image attributes like style and genre are critical for performance, while some computed features may be redundant due to their extraction by deep network backbones. An analysis of failure cases underscores the challenges in predicting subjective aesthetic judgments and the issue of overfitting to user-specific patterns.

**Strengths**
- Novelty: First dataset tailored for PIAA in the context of artistic images, addressing limitations of previous datasets that focused on natural images.
- Rich Annotations: Combines both detailed image attributes and extensive personal annotator information, enabling more nuanced aesthetic assessments.
- Rigorous Curation: Involvement of art historians in the data curation process enhances the dataset’s credibility and relevance.
- Comprehensive Experiments: Implementation and evaluation of both GIAA and PIAA models provide a thorough empirical foundation.
- Ablation Studies: Systematic analysis of the contribution of different attributes to model performance.
- Insightful Analysis: Detailed examination of failure cases and model limitations, particularly regarding subjective ratings and overfitting issues.

**Minor note.** I was tempted to rate it as "borderline" as I feel the work could get more visibility in other venues. However, (1) the quality and diversity of this submission will speak well about the CVEU workshop, and (2) it would be arrogant to impose my opinion on the authors' intention.

**Departing note.** If accepted, I strongly encourage the authors to engage and get feedback from other attendees of the workshop. Please prepare multiple presentations of your work (elevator pitch, lunch/snack, ..., poster/talk) such that you can break the ice, and make the most of the event :).

---

### Official Review · Reviewer_TbL7 · 2025-03-15
**A novel benchmark**

**Rating:** 4
**Confidence:** 4

**Review:**

This paper introduces LAPIS, the first dataset for personalized image aesthetic assessment (PIAA) in artworks, consisting of 11,723 curated images with aesthetic scores, image attributes, and personal annotator attributes.

Collaborating with art historians, the authors ensure high-quality annotations and evaluate two state-of-the-art PIAA models, conducting ablation studies to show the importance of personal and image attributes.

The dataset has strong applications in art creation and AI-driven aesthetics, but potential biases in annotator diversity and the lack of a new modeling approach limit its broader impact.

---

### Official Review · Reviewer_NZrA · 2025-03-22
**Great contribution for computational aesthetics on art**

**Rating:** 5
**Confidence:** 3

**Review:**

This paper introduces LAPIS, a high-quality dataset for personalized image aesthetic assessment (PIAA) in the domain of art. The dataset includes 11,723 carefully curated images annotated by over 500 participants, with both image- and annotator-level metadata. The authors also run extensive analysis and evaluate two prior PIAA models on the new dataset.

# Pros
- LAPIS fills a clear gap in the field: there is currently no dataset for PIAA focused on artworks. I'm quite excited about what this can enable for future artistic image generation.
- The data curation process appears thoughtful and rigorous, including manual cleaning and art-historian-guided label refinement. The inclusion of art historians in the process is great.
- The dataset includes rich metadata on both images and annotators, which is crucial for personalization tasks.
- The paper includes in-depth analysis, ablations, and model evaluation using prior state-of-the-art methods.

# Cons
- Some parts of the draft are in need of proofreading (e.g., missing periods, inconsistent phrasing), though this is minor.

# Final Remarks
I believe this paper is a valuable dataset contribution and would be a strong fit for a CVPR workshop. While not methodologically novel, it provides a well-documented resource for a growing area of interest in computational aesthetics.

---

### Official Review · Reviewer_Ld4N · 2025-03-24
**Recommendations and Review for Enhancing the Clarity and Rigor of "LAPIS: A novel dataset for personalized image aesthetic assessment"**

**Rating:** 4
**Confidence:** 3

**Review:**

**Overall Assessment:**
* Work proposes a dataset addressing problems in PIAA datasets. The proposed dataset is mainly similar to "PARA:Personalized Image Aesthetics Assessment with Rich Attributes" paper but focuses on mainly art pieces making it more specific. Work is not as detailed and extensive as mentioned PARA paper however it is clear in general. There are parts which can be improved for clarity.
*This can be accepted with a rating of 4 weak accept.

* **Quality:** The work presented appears to be of good quality overall.
* **Clarity:** While the paper is generally understandable, there are areas where clarity could be improved.
* **Originality:** The work demonstrates some originality.
* **Significance:** The research has some significance.

**Pros:**

* The paper's proposal of a novel dataset for the task with detailed image attributes represents a contribution to the literature, holding some significance for the field.

**Cons:**

* Some parts and fıgures can be more clear and some parts are not clearly stated as in supmat.


**Possible Improvements**
* The selection of balanced aspect ratio images (line 226) may introduce bias, and the paper would benefit from elaborating on potential impacts and considering if certain styles are underrepresented. This could impact the overall clarity of the study's scope.
* Figure 2's aesthetic score (0-100) has only 7 underlying values (supmat), which would benefit from clarification in the main text to avoid misinterpretation of granularity and improve clarity.
* The paper lacks differentiation by artist or time period, potentially biasing the dataset due to uneven sample distribution. This potential limitation warrants acknowledgement for improved clarity and understanding of the dataset's characteristics.
* Overlapping data points in Figure 4 obscure correlation significance, impacting the clarity of the findings. Alternative visualization techniques such as transparency or density plots could be considered or the plot can be improved.
* Figure 6 is missing the "slightly aesthetic" label, which affects the clarity and completeness of the annotation representation. The inclusion of this label would provide a complete representation of annotations.
* Figure 7 lacks metric results for ResNet50 The addition of these would improve the clarity of the results section.
* The GAN used for upscaling (supmat) was trained on abstract art. Its application across all genres may introduce unwanted results, potentially affecting the quality of the image data used and the clarity of the methodology. Justification for this choice or exploration of other SRR techniques is recommended.

---

### Decision · Program_Chairs · 2025-03-25

**Decision:**

Accept

**Comment:**

The paper introduces LAPIS, a carefully curated dataset designed specifically for personalized image aesthetic assessment (PIAA) in the domain of artworks, comprising 11,723 images annotated with extensive metadata and personal attributes. Reviewers unanimously praised its novelty, rigorous data curation involving art historians, comprehensive annotations, and insightful ablation studies. Minor concerns regarding clarity, potential biases, and granularity in figures were noted but do not significantly detract from the dataset's overall value.

Given the clear strengths, relevance, and overall positive reception, the paper is clearly accepted. The authors are encouraged to address reviewers' comments in the camera-ready submission by improving figure clarity, elaborating on potential biases, and refining dataset descriptions for better readability.